# Cryo-EM Structure of the Flagellar Motor Complex from *Paenibacillus* sp. TCA20

**DOI:** 10.3390/biom15030435

**Published:** 2025-03-18

**Authors:** Sakura Onoe, Tatsuro Nishikino, Miki Kinoshita, Norihiro Takekawa, Tohru Minamino, Katsumi Imada, Keiichi Namba, Jun-ichi Kishikawa, Takayuki Kato

**Affiliations:** 1Institute for Protein Research, Osaka University, Suita 565-0871, Osaka, Japannishikino.tatsuro@nitech.ac.jp (T.N.); 2Department of Life Science and Applied Chemistry, Nagoya Institute of Technology, Nagoya 466-8555, Aichi, Japan; 3OptoBioTechnology Research Center, Nagoya Institute of Technology, Nagoya 466-8555, Aichi, Japan; 4Graduate School of Frontier Biosciences, Osaka University, Suita 565-0871, Osaka, Japannamba.keiichi.fbs@osaka-u.ac.jp (K.N.); 5JEOL YOKOGUSHI Research Alliance Laboratories, Osaka University, Suita 565-0871, Osaka, Japan; 6Department of Macromolecular Science, Graduate School of Science, Osaka University, Toyonaka 560-0043, Osaka, Japan; 7Faculty of Applied Biology, Kyoto Institute of Technology, Kyoto 606-8585, Kyoto, Japan

**Keywords:** flagella motor, MotA1/MotB1 complex, cryo-EM, SPA, stator complex, antibiotics

## Abstract

The bacterial flagellum, a complex nanomachine composed of numerous proteins, is utilized by bacteria for swimming in various environments and plays a crucial role in their survival and infection. The flagellar motor is composed of a rotor and stator complexes, with each stator unit functioning as an ion channel that converts flow from outside of cell membrane into rotational motion. *Paenibacillus* sp. TCA20 was discovered in a hot spring, and a structural analysis was conducted on the stator complex using cryo-electron microscopy to elucidate its function. Two of the three structures (Classes 1 and 3) were found to have structural properties typical for other stator complexes. In contrast, in Class 2 structures, the pentamer ring of the A subunits forms a C-shape, with lauryl maltose neopentyl glycol (LMNG) bound to the periplasmic side of the interface between the A and B subunits. This interface is conserved in all stator complexes, suggesting that hydrophobic ligands and lipids can bind to this interface, a feature that could potentially be utilized in the development of novel antibiotics aimed at regulating cell motility and infection.

## 1. Introduction

Bacterial flagella are motility organelles, complex nanomachines composed of more than 30 kinds of proteins, with copy numbers ranging from a few to tens of thousands [1,2,3]. Many motile bacteria use flagella to swim in liquid environments and move on solid surfaces, advancing toward favorable environments for growth and infection. Therefore, flagella play a crucial role in the survival of bacteria.

The bacterial flagellum consists of the rotor complex surrounded by multiple stator complexes. The rotor complex is composed of a basal body embedded within the cell envelope, from which a long filament extends. The stator complexes act as ion channels, converting the electrochemical potential of ions across the membrane into the torque of the rotor complex.

Two types of membrane proteins, A and B subunits, form a stator complex. *Escherichia coli* (*Ec*) and *Salmonella enterica* (*Se*) [4] have a H^+^-driven stator whose A and B subunits are named MotA and MotB, while in *Vibrio* species such as *V. cholerae* or *V. alginolyticus* (*Va*), the PomA and PomB subunits form a Na^+^-driven stator [5]. In some *Bacillus* species such as *Bacillus subtilis* (*Bs*), the Na^+^-driven stator proteins are referred to as MotP and MotS [6]. Based on a cryo-electron microscopy analysis, the structures of the stator complex have been determined at near-atomic resolution for several bacterial species, including *Campylobacter jejuni* (*Cj*), *B. subtilis*, *Clostridium sporogenes* (*Cs*), and *V. alginolyticus* (*Va*) [7,8,9,10,11]. These structures share a common feature: a ring composed of five A subunits with two B subunits penetrating the center of the A-subunit pentamer ring (A_5_B_2_).

The A subunit contains four transmembrane helices and a cytoplasmic domain [12]. The highly conserved charged residues in the cytoplasmic domain of the A subunit are important for interacting with the cytoplasmic rotor protein FliG to generate torque for motor rotation [3,13,14,15,16]. The B subunit consists of a single transmembrane helix in the N-terminal region and a periplasmic OmpA-like domain in the C-terminal region, connected by a long linker. A conserved aspartic acid residue in the transmembrane helix of the B subunit is essential for ion translocation across the cytoplasmic membrane [17]. The OmpA-like domain binds to the peptidoglycan layer, anchoring the stator around the rotor [2,3]. The linker region contains a short helix called the plug helix, which controls the ion flux through the stator complex [3,18,19,20,21,22]. Ion transport does not occur when the stator complex is inactive, due to the interaction between the plug helix and the periplasmic loop of the A subunit.

Upon assembly of the stator complex around the rotor, the interaction between the A subunit and FliG may induce conformational change of the B subunit long linker region, which triggers the dissociation of the plug helix from the A subunits. As a result, the OmpA-like domain binds to the peptidoglycan layer, allowing ion flux in the stator complex. Based on this structural information, a torque generation model was proposed, where clockwise rotation of the A-subunit ring against the B-subunit dimer is coupled to ion flux, which powers rotation of the basal body, although rotation of the A-subunit ring has not yet been demonstrated.

*Paenibacillus* sp. TCA20 (*Pb*) was discovered in a calcium-rich hot spring, and the MotA1/MotB1 complex was reported to be driven by the divalent cation Ca^2+^ [23]. However, in 2020, Onoe et al. demonstrated that *Pb*MotA1/MotB1 functions as a H^+^-driven stator by using a chimeric MotB1 protein in which the MotB1 C-terminal domain is replaced with the corresponding part of the *E. coli* MotB protein [24]. In this study, we purified the chimeric MotA1/MotB1 complex from the *E. coli* expression system without flagellar formation, performed its structural analysis by using cryo-electron microscopy (cryo-EM) single particle analysis, and present here three structures of the chimeric MotA1/MotB1 complex in the inactive state. Furthermore, we show that a detergent molecule is inserted into the inner side of the MotA1 ring, thereby significantly distorting the symmetry of the MotA1 ring from C5.

## 2. Materials and Methods

### 2.1. Bacterial Strain and Plasmids

*E. coli* cells were cultured in LB medium (Lennox, Nacalai Tesque, Kyoto, Japan). Ampicillin and chloramphenicol were added at final concentrations of 50 and 25 μg mL^−1^, respectively. To construct a plasmid for expression of the chimeric MotA1/MotB1 complex, named pColdIV-chimeraric_motA1motB1-his_6_, the *motA1* and chimeric *motB1* genes were amplified by PCR using a plasmid (pSHU165) [24] as a template, a forward primer (f_pBAD24MotA1B1), and a reverse primer (r_pBAD24MotA1B1). The vector fragment was amplified by PCR using the pColdIV-pomApomB-his6 [25] plasmid as a template, a forward primer (f_pCold4pomAB_His6), and a reverse primer (r_pCold4pomAB_His6), and the DNA fragment encoding the *motA1* and chimeric *motB1* genes was then inserted into the vector fragment via In-Fusion cloning (Takara Bio Inc, Kusatsu, Japan) (Appendix A). *E. coli* transformation was performed using a standard method with CaCl_2_.

### 2.2. Sample Preparation

Purification of the chimeric MotA1/MotB1 stator complex was performed as previously described with some modifications [25]. *E. coli* BL21(DE3)/pLysS cells carrying the pColdIV-chimeraric_motA1motB1-his_6_ were grown overnight in 20 mL of LB medium at 37 °C, inoculated in 2 L of LB medium, and cultured at 37 °C until an optical density of 0.5 was reached at 660 nm. After incubation on ice for 30 min, IPTG was added at a final concentration of 0.5 mM, followed by culturing for 1 day at 15 °C. The cells were collected by centrifugation at 8000× *g* for 10 min and suspended in 7 mL of Na-Pi buffer (50 mM sodium phosphate (pH 8.0), 200 mM NaCl, and 10% (*w*/*v*) glycerol) per 1 g (wet weight) of the cells. The cells were disrupted by ultrasonication (Astrason XL-2020). After removal of unbroken cells by low-speed centrifugation at 8000× *g* for 10 min, the cell lysate was ultracentrifuged at 118,000× *g* for 1 h. The pellet was suspended in the original volume of Na-Pi buffer and stored at −30 °C until use.

The frozen suspension was thawed in a water bath, and the cell membranes were solubilized using lauryl maltose neopentyl glycol (2,2-didecylpropane-1,3-bis-β-D-maltopyranoside, LMNG) at a final concentration of 0.5% (*w*/*v*) with stirring for at least 60 min at 4 °C. Insoluble membranes were removed by centrifugation at 120,000× *g* for 30 min. The supernatant was mixed with Ni-NTA agarose resin (QIAGEN, Hilden, Germany) equilibrated in wash buffer (20 mM Tris-HCl, pH 8.0, 100 mM NaCl, 20 mM imidazole, and 0.05% (*w*/*v*) LMNG), and the mixture was stirred for at least 1 h at 4 °C. The column volume was determined according to the total volume of resin in the column. After mixing, the mixture was poured into a polypropylene column, the supernatant was eluted from the column, and the His-tagged stator complex was purified using a batch method. Wash buffer was added to wash the column and then removed. To elute the His-tagged stator from the resin, two volumes of elution buffer (20 mM Tris-HCl, pH 8.0, 100 mM NaCl, 200 mM imidazole, and 0.05% (*w*/*v*) LMNG) was added. The peak fractions were collected and concentrated using an Amicon device with 100 kDa cutoff (Millipore, Burlington, MA, USA). The concentration of the purified sample was measured based on 280 nm absorption using a NanoDrop (Thermo Fisher Scientific, Waltham, MA, USA). This sample was also reconstituted into peptidiscs. The peptidisc solution was prepared from bulk lyophilized peptidiscs (Peptidisc Biotech, Vancouver, BC, Canada) dissolved in 20 mM Tris-HCl, pH 8.0, to a final concentration of 2 mg mL^−1^. The dissolved peptidisc solution was mixed with the solubilized stator complex sample at a 1:1 weight ratio and incubated at room temperature for 30 min. The mixture was loaded on a size-exclusion column (Superdex200 Increase 10/300 GL column, GE Healthcare, Milwaukee, WI, USA) equilibrated with buffer C (20 mM Tris-HCl, pH 8.0, 100 mM NaCl) and eluted with buffer C at a flow rate of 0.5 mL min^−1^. The stator complex in detergent-free solution was concentrated using an Amicon device with a 100 kDa cutoff (Millipore). The concentration of the purified sample was measured based on 280 nm absorption using a NanoDrop (Thermo Fisher Scientific). The sample purity was assessed using sodium dodecyl sulfate polyacrylamide gel electrophoresis (SDS-PAGE) with Coomassie Brilliant Blue R250 (CBB) staining.

### 2.3. Cryo-EM Acquisition

Purified chimeric MotA1/MotB1-His6 (10 µL of ~1.5 mg mL^−1^ solution) was loaded onto a Quantifoil holey carbon grid R1.2/1.3 Cu 300 mesh (Quantifoil Micro Tools GmbH, Großlöbichau, Germany) with pretreatment of one side of the grid by glow discharge. After 3 min incubation at room temperature, 2.7 µL of freshly purified protein was loaded onto the grids again, which were blotted and plunged into liquid ethane for rapid freezing using Vitrobot Mark IV (Thermo Fisher Scientific) with a blotting time of 4–7 s at 4 °C and 100% humidity.

The cryo-EM movies were recorded using Titan Krios (FEI) equipped with a thermal field-emission electron gun operated at 300 kV, an energy filter with a 20 eV slit width, and a bioquantum K3 direct electron detector camera (Gatan, Pleasanton, CA, USA). The cryo-EM data were automatically collected by the SerialEM software v3.7 [26]. The dose-fractionated movies were recorded at a nominal magnification of 81,000×, corresponding to an image pixel size of 0.88 Å. The total electron dose was 50 e^−^/Å^2^. The collected spherical abbreviation was 0.073 mm. A total of 3129 movies were recorded.

### 2.4. Image Processing

All image processing was performed using cryoSPARC v3.8 [27]. The workflows are summarized in Appendix A. Motion correction and CTF estimation were applied to the recorded movies. Based on manual inspection and curation, 3048 micrographs were used. A blob particle picker was used for initial particle picking in the complex. The picked particles were subjected to three rounds of 2D classification. High-quality 2D average images were selected for template-based particle picking using Topaz v 0.25 [28]. After several rounds of 2D classification, 168,628 carefully selected particles were identified from a total of 765,007 particles for building an initial model via ab initio reconstruction. Heterogeneous refinement was performed to classify the particles into six classes, with further homogeneous refinement on three classes with well-defined structures. This refinement resulted in separate density maps for the chimeric MotA1/MotB1, with resolutions of 3.3 Å, 3.3 Å, and 3.5 Å, respectively.

### 2.5. Model Building

To build the atomic model of the chimeric MotA1/MotB1, the atomic model predicted by AlphaFold2 was used as the initial model [29]. This model was fitted into the density map as a rigid body and then manually refined using Coot software v0.89 [30]. The manually corrected model was refined using Servalcat software 0.2 [31]. This iterative process was performed for several rounds to correct the remaining errors until the model geometry was satisfactory. The model quality was assessed based on Molprobity scores [32]. The statistics of the maps and models presented here are summarized in Appendix A. All figures were prepared using UCSF ChimeraX software v1.8 [33].

## 3. Results

### 3.1. Structural Analysis of the MotA1/MotB1 Complex

A structural analysis of the MotA1/MotB1 complex was carried out using a chimeric protein comprising the C-terminal periplasmic domain of *E. coli* MotB and the transmembrane domain of MotB1 from *Paenibacillus* sp. TCA20, hereafter referred to as *Pb*MotB1. Although the WT *Pb*MotA1/MotB1 was not functional within *E. coli* cells, the chimeric complex has been confirmed to be functional in *E. coli* [24]. Therefore, functional analyses can be easily conducted in *E. coli*. The chimeric MotA1/MotB1 complex was expressed in *E. coli*, solubilized using LMNG detergent, and purified by Ni-affinity chromatography and then size exclusion chromatography (SEC). After reconstitution of this complex into a peptidisc formed by short amphipathic bi-helical peptides [34], the sample was again run on SEC to remove free peptides. The sample was quickly frozen on a grid, a dataset of 3129 cryo-EM movies was collected, and the three-dimensional (3D) structure was analyzed using a single particle analysis.

As a result of 3D classifications, three distinct structures of the complex, Classes 1, 2, and 3, were obtained at resolutions of 3.3 Å, 3.3 Å, and 3.5 Å, respectively (Figure 1a). They share the same basic structure, with differences observed in the distance between the MotA1 protomers and the MotB1 structure. These differences result from flexibility and external factors, such as the binding of detergent (details discussed in Section 3.3). Each structure is represented by a heteroheptamer composed of five MotA1 and two chimeric MotB1 molecules (A_5_B_2_) and chains A to G (Figure 1c), having the same stoichiometry and topology as previously reported for the structures of stator complexes from other bacterial species [7,8,9,11].

The A subunit consists of four transmembrane helices (α1, α2, α7, and α8) and five cytoplasmic helices (α4–α9) (Figure 1b). The two transmembrane helices of the two B subunits are tightly paired and penetrate into a pentameric ring formed by MotA1, and a short single helix from each B subunit, called the plug helix, is bound to the periplasmic surface of the A subunit ring. The C-terminal region of the chimeric *Pb*MotB1 analyzed here is derived from *E. coli* MotB (residues 104–302). The MotB1 region visualized by cryoEM extends up to residue 60, corresponding to the native part of the MotA1/MotB1 complex from *Paenibacillus* sp. TCA20.

### 3.2. Mechanism of Rotor Rotation by MotA1 in E. coli

In this study, we analyzed the B subunit of the MotA1/MotB1 complex, a chimeric protein composed of the N-terminal domain of *Pb*MotB1, which directly interacts with *Pb*MotA1, and the C-terminal domain of *Ec*MotB, which interacts with the peptidoglycan layer of the cell. It has already been confirmed that the stator complex formed by *Pb*MotA1 and this chimeric MotB1 is functional in rotating the *E. coli* rotor in vivo [24]. In other words, the *E. coli* rotor protein FliG interacts with *Pb*MotA1 of the chimeric stator complex to generate motor torque.

The A subunit ring is thought to rotate relative to the B subunit anchored to the peptidoglycan, in conjunction with ion influx across the membrane. This rotation is transmitted to the flagellar rotor, causing the flagellum to rotate. Therefore, the interaction between *Pb*MotA1 and EcFliG is crucial for torque generation. In 2024, Johnson et al. succeeded in determining the structure of the MotA-FliG complex in *C. sporogenes* and revealed that the singular C-terminal domain of FliG interacts with two MotA subunits, where the lower portions of α5 and α6 of a *Cs*MotA subunit, together with α4 and α9 of the neighboring MotA subunit, are involved in the MotA–FliG interaction [35]. When the structure of the analyzed *Pb*MotA1 was compared with *Cs*MotA, the root mean square deviation (RMSD) was 2.11 Å, indicating relatively high structural similarity. This allows for predictions of the interaction sites between *Pb*MotA and *Ec*FliG. In two MotA1 subunits, we identified nine residues (R89, R90, V93, L94, E97, Q113, D117, N233, and L235) within 5 Å of FliG that can potentially interact with it. A sequence comparison with *Ec*MotA revealed that R89 and E97 in *Pb*MotA1 are conserved, while the residues R90, V93, L94, and D117 have similar properties. On the other hand, Q113, N233, and L235 showed no sequence similarity (Figure 2).

*Vibrio alginolyticus* PomA (*Va*PomA) is the A subunit of the Na^+^-driven stator that can rotate the *E. coli* rotor in a way similar to *Pb*MotA1. In *V. alginolyticus*, *E. coli*, and *Salmonella enterica*, conserved charged residues within positions 89 to 100 in PomA or MotA are important for interactions of PomA or MotA with FliG [3,13,14,15,16]. *Pb*MotA1 showed a higher sequence similarity to *Vp*PomA than to *Ec*MotA. The residues in *Pb*MotA1 predicted to interact with *Ec*FliG, R89, R90, and E97, correspond to highly homologous residues among the three species, suggesting that *Pb*MotA1 can also interact with *Ec*FliG.

### 3.3. Structural Comparison of the Three MotA1 Rings

As described above, three different structures of *Pb*MotA1/MotB1 were obtained from image processing (Figure 1a). These structures were almost identical and in an inactive state, with the plug helices of MotB1 located on the periplasmic surface of MotA1. However, several differences were observed, particularly in the *Pb*MotA1 ring shapes and the plug helix of *Pb*MotB1.

The Class 1 structure exhibits high similarity to the *Cj*MotA ring of previously reported stator complexes (Appendix A). There is a cavity between chains D and E of the *Pb*MotA1 ring and chain F of MotB1 (Figure 3a), allowing solvent to access, from both the periplasmic and cytoplasmic sides, essential residues involved in H^+^ binding (Asp24). In contrast, another cavity between *Pb*MotB1 (chain G) and *Pb*MotA1 (chains B and C) was not open to the solvent, hindering access of H^+^ and its binding to residue Asp24 (Figure 3a). These features closely match those of previously reported stator complexes [7,8,9,10,12], suggesting that this structure represents an inactive form that diffuses around the membrane before its incorporation into the motor.

Class 2 had the largest population of particle images among the three classes. In the Class 1 structure, the MotA1 ring has pseudo-pentagonal symmetry. In contrast, in the Class 2 structure, there is a large cleft between two neighboring subunits of MotA1, namely chains A and E (Figure 3b). Within this region, no direct interaction between these two subunits was found on either the periplasmic or cytoplasmic sides. When comparing the conformations of all five subunits, the RMSD was less than 1.0 Å (Appendix A), indicating there were no significant structural differences. Therefore, this large conformational change in the MotA1 ring seems to be caused by subtle differences in the interface between the subunits.

Additionally, in Class 1 and chains B to E in Class 2, the α1 TM helix interacted with the α2 TM helix of neighboring subunits. However, in the Class 2 structure, no α1 TM helix of chain A was observed. This could be due to the large cleft between the pair of neighboring MotA1 subunits, chains A and E, which precludes interaction. While this structure is expected to be unstable due to the ring cleft and resulting loss of interaction, the majority of particles were classified as Class 2, and a high-resolution structural analysis was successfully achieved. These findings suggest that this conformation is not a transient structure.

In Class 2, the detergent LMNG used for solubilization was found between chains C and D of MotA1 and chain G of MotB1 (Figure 3c). The LMNG molecule was bound to its two alkyl chains oriented toward the transmembrane region of MotA1/MotB1, while its hydrophilic portion was exposed to the periplasmic space (Figure 3c). As a result, the α6 helix of chain C was pushed outward by approximately 7.4 Å, and chain D was shifted approximately 13 Å toward chain C. This shift propagated to the next chain E, which was displaced approximately 10 Å toward chain D. However, this significant displacement did not strongly propagate to chain A, where the shift was only approximately 5 Å (Figure 3d). The difference in the propagation of these structural shifts likely resulted in formation of the large cleft between chains A and E (Figure 3b).

The Class 3 structure showed a high similarity to that of Class 1, with an RMSD of 0.53 Å. In contrast to Class 2, neither a large cleft nor significant deformation was observed within the MotA1 ring. LMNG was also absent. Despite high structural similarities to Class 1, three of the α1 TM helices of MotA1 (chains A, B, and C) were not observed (Figure 3b). Based on the results from Class 2, it was expected that LMNG binding would cause significant distortions in the ring array of subunits, which would hinder visualization of the α1 TM helices. However, the structure of Class 3 suggests that the interaction between the α1 TM helix of a MotA1 subunit and the α2 TM helix of the neighboring subunit is not essential for ring formation, which would occur in a state of equilibrium between association and dissociation.

### 3.4. The Plug Helix of MotB1

The plug helix has been shown to regulate ion permeability [3,18,19,20,21,22,23]. In the structures of Classes 1 and 3, the two plug helices of the two MotB1 subunits (chains F and G) were clearly visible and were located at the periplasmic surface between two neighboring MotA1 subunits (between chains A and B and between chains F and G). Furthermore, the plug helix bound to opposing sides of the MotA1 ring, by crossing the linker to their connecting TM helices (Figure 4).

On the other hand, only one plug helix was observed in the Class 2 structure, and the other was not visible (Figure 4). This is the first report of this unique stator complex structure, given that in all previously reported structures, both plug helices are bound to the ring of A subunits [7,8,9,11]. As described above, LMNG was bound to the MotA1 ring in the Class 2 structure. The position of the LMNG hydrophilic head partially overlapped with the region of the plug helix for binding onto MotA1, suggesting that the binding of LMNG inhibited the plug helix from binding to MotA1. This is consistent with the structure of Class 2, in which one of the two plug helices was not visualized.

It is known that when the plug helix is absent, the ion channel remains open. In the Class 2 structure, although one of the two plug helices is not bound to MotA1, the cavity on the periplasmic side is closed, thereby preventing ions and solvents from accessing the ion binding site. Additionally, LMNG bound in place of the plug helix appears to be obstructing ion permeation. Collectively, these results indicate that the Class 2 ion channel is structurally in an inhibited state even when one of the two plug helices is unbound.

## 4. Discussion

In this study, we performed a structural analysis of the chimeric stator complex consisting of MotA1 and MotB1 from *Paenibacillus* sp. TCA20 with the C-terminal region of MotB1 replaced with the corresponding region of *Ec*MotB. Our results revealed the structure of the chimeric MotA1B1 complex, although the C-terminal region of the chimeric B subunit after the plug helix could not be visualized. From this analysis, we visualized a unique stator complex structure that differs from any of those previously reported. Of the three classified structures we determined, the structure of Class 2 exhibited a significantly more open and distorted ring conformation, where direct contact between chains A and E of MotA1 was lost. Additionally, one of the two plug helices of MotB1 was not visualized, and the other plug helix along with its linker connecting to the TM helix adopted a different conformation compared to the other two classes in which the two plug helices were stably bound to the periplasmic surface of the MotA1 pentamer ring. The open and distorted ring structure of Class 2 appears to be structurally unstable; however, the specimen we analyzed by cryo-EM exhibited high thermal stability, and this structure is the most highly represented in our cryo-EM dataset.

The Class 2 structure was stabilized by the presence of a detergent molecule, LMNG, which directly binds to the MotA1 ring. Even after replacing the detergent micelle with a peptidisc, LMNG remains stably bound. This is the first structure of a H^+^-driven stator complex where a small molecule bound in the complex perturbs the conformation. Given the significant structural deviation from previously analyzed stator complexes, it is likely in an inactive state. In this LMNG-bound structure (Class 2), the significant distortion of the MotA1 ring strongly suggests that the smooth rotation of MotA1 relative to MotB1 is inhibited. Additionally, one of the plug helices is displaced, with LMNG bound in its place, which implies that ion influx is also inhibited. Therefore, small molecules that mimic this binding pattern could serve as potential candidate antibiotics. LMNG interacts with numerous hydrophobic residues in the loop between helices α7 and α8 (residues 165–182) of MotA1 and residues 34–52 of MotB1 (Appendix A). These residues are highly conserved among many bacteria. Therefore, a novel antibiotic that mimics this binding has potential as a broad-spectrum antibiotic.

Recently, the structure of the Na^+^-driven *V*aPomA/PomB complex was analyzed as a complex with the Na^+^ channel blocker phenamil [7]. Phenamil was found to bind to the interface for cytoplasmic interaction between PomA and PomB, suggesting that phenamil binding to the stator may block the PomA ring rotation coupled with sodium ion flux. When we compared the phenamil binding site of *Vibiro* PomA/PomB with the corresponding part of the chimeric MotA1MotB1 stator, few residues were conserved (Appendix A), suggesting that phenamil might be ineffective at inhibiting the *Pb*MotA1/MotB1 stator. The binding position of LMNG found in this study is completely different from the binding position of the phenamil. The structural changes induced by LMNG binding are thought to directly inhibit both MotA1 ring rotation and ion influx, indicating a distinct mechanism of inhibition. Additionally, while molecules that mimic phenamil binding act as Na^+^ channel inhibitors and specifically target Na^+^-driven flagellar motors, molecules that mimic LMNG binding interact with highly conserved hydrophobic regions of both the A and B subunits. There is a technical hurdle in that an in vitro assay for the measurement of the coupling ion flux of the stator has not been established yet, and this hurdle will be solved in a further study. But, by combining such a new assay with the screening of small molecules designed based on the structure of the LMNG, new drugs may be developed to serve as antibiotics with a much broader antimicrobial spectrum than those presently available.

Additionally, extra densities were found near the α7 helix of MotA1, but they were not uniformly distributed across the five α7 helices of the five MotA1 proteins (Appendix A). Because these densities were too small to assign any amino acid residues, possible candidates include the unassigned C-terminus of MotA1, peptides other than MotA1/MotB1, lipids, or detergents, but further validation is required to elucidate the identity.

## 5. Conclusions

In this study, we analyzed the structure of the MotA1/MotB1 stator complex derived from *Paenibacillus* sp. TCA20 by cryo-EM image analysis and successfully visualized three structures, including a unique structure distinct from any of the previously analyzed structures. In this structure, LMNG binds between the A and B subunits, causing one part of the ring formed by the five A subunits to completely dissociate, which results in a C-shaped structure. Furthermore, LMNG binding inhibited the interaction of the plug helix of the B subunit with the A subunit. These structural changes are predicted to disrupt the function of the H^+^ channel. This result represents the first structure of the chimeric stator with a small molecule bound to a H^+^-driven stator complex. Together with the structural information of other H^+^- and Na^+^-driven stator complexes, the present structures of the chimeric MotA1/MotB1 stator help us understand ion selectivity and energy conversion mechanisms of molecular motors and are expected to contribute to the development of novel antibiotics targeting the flagellar motor.

## Figures and Tables

**Figure 1 biomolecules-15-00435-f001:**
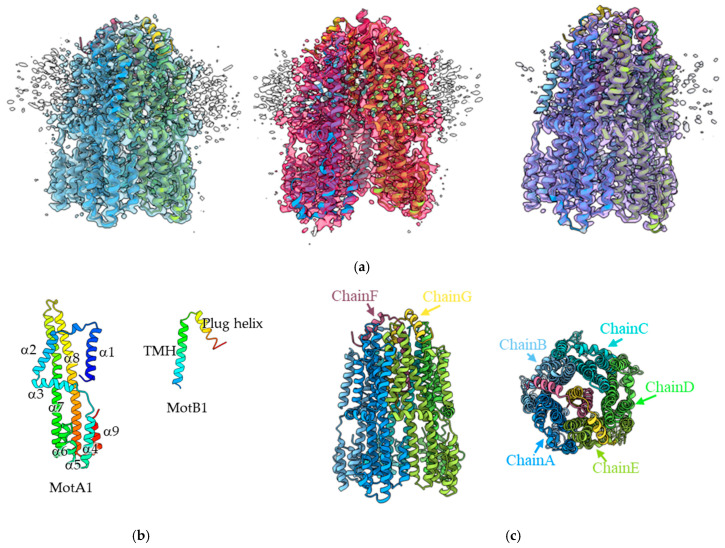
Three-dimensional structure of the *Pb*MotA1/MotB1 complex. (**a**) Structures of the three different classes. The ribbon models for Classes 1, 2, and 3 are shown to the left, middle, and right, respectively. Each EM map is superimposed on the model. (**b**) Topology of the A and B subunits. TMH, transmembrane helix. It is rainbow colored; blue (N-teminus) to red (C-terminus). (**c**) Side and top (from periplasm) views of Class 1 are shown. Chains A–E and chains F and G are MotA and MotB protomers, respectively.

**Figure 2 biomolecules-15-00435-f002:**
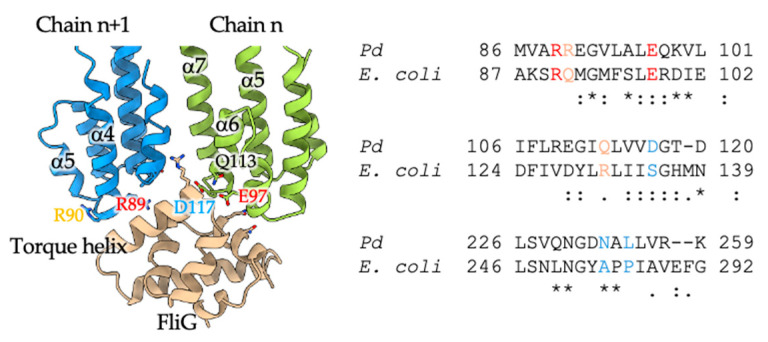
Interaction between MotA1 and FliG: A ribbon diagram of the MotA-Fig complex indicating residues involved in the interaction (left). The two MotA1 subunits are colored blue or green, and FliG is in light brown. The sequence homology of the interaction residues between *Pb*MotA1, *Ec*MotA, and *Vp*PomA is shown on the right. The interaction residues are color-coded as follows: red represents conserved residues (asterisk), orange indicates residues with similar properties (colon), and blue denotes different properties. Some interacting residues in *Pb*MotA1 are labeled. The structure of *Ec*FliG was predicted using AlphaFold2.

**Figure 3 biomolecules-15-00435-f003:**
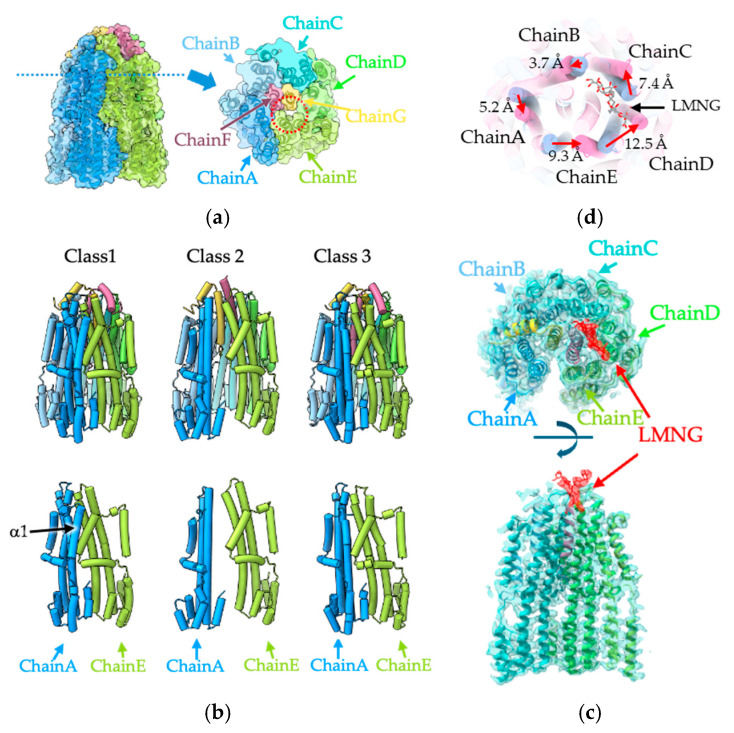
Structural comparison of Classes 1–3. (**a**) Surface rendering of Class 1 (left) and the cross-section at the blue dotted line in the left panel (right). The red dotted circle indicates the gap near the D24 region of chimeric MotB1. The chain IDs correspond to those in Figure 1c. (**b**) Structural comparison of the three structures using tube rendering of α-helices. The upper panel shows all chains, and the lower panel shows only chain A (blue) and chain E (yellow green). (**c**) Top (upper) and side (lower) views of the Class 2 structure are shown in the ribbon model superimposed on the EM map. Specific binding of LMNG (red) is shown in the stick model. (**d**) The positional shift in each subunit induced by LMNG binding: blue tubes represent Class 1, and pink tubes represent Class 2, with only the α8 TM helices emphasized; the rest are transparent. The two structures are aligned with the chimeric MotB1 chain G.

**Figure 4 biomolecules-15-00435-f004:**
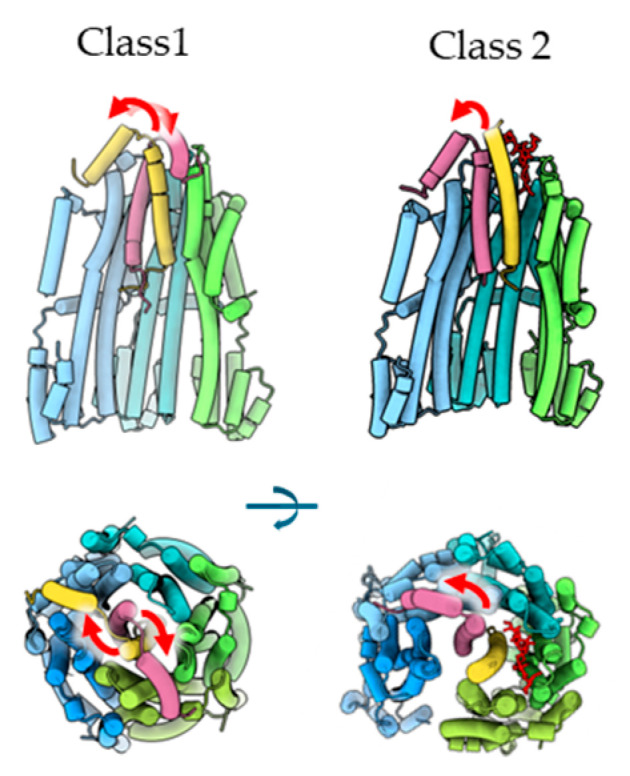
Differences in the plug helices of MotB1 between Classes 1 and 2: The upper panel shows the side view, and the lower shows the top view. In the side view, chains A and E (light green and blue) were removed to see the positions of MotB1 subunits. The red arrows indicate the direction of the connection between the TM helix and the plug helix.

## Data Availability

All the data obtained in this study are available in the main text, the Appendix A, and the Appendix A. The atomic coordinate has been deposited in Protein Data Bank, www.rcsb.org, under accession codes 9LUB (*Pb*MotA1/MotB1, Class 1), 9LU9 (*Pb*MotA1/MotB1, Class 2), and 9LUC (*Pb*MotA1/MotB1, Class 3). The cryo-EM maps have been deposited in the Electron Microscopy Data Bank under accession codes EMD-63393 (*Pb*MotA1/MotB1, Class 1), EMD-63392 (*Pb*MotA1/MotB1, Class 2), and EMD-63394 (*Pb*MotA1/MotB1, Class 3).

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
