# Peer review of "Cryo-EM Structure of the Flagellar Motor Complex from Paenibacillus sp. TCA20"

_biomolecules, 2025, doi:10.3390/biom15030435_

Round 1

Reviewer 1 Report

Comments and Suggestions for Authors

The manuscript provides a detailed and comprehensive structural analysis of the chimeric MotA1/MotB1 complex from Paenibacillus sp. TCA20 using cryo-EM. While the overall structure is well-presented, certain sections are densely packed with technical jargon and could benefit from additional simplification or explanation to make them more accessible, especially for readers outside of the structural biology field.

General comments

  1. The study is highly relevant and adds valuable new insights into the function of bacterial flagellar motors, particularly those driven by calcium ions. The use of cryo-EM to capture the structure of the stator complex in a detergent-free state (peptidisc) is a significant contribution. However, the implications of these findings for antibiotic development, while touched upon in the discussion, could be more thoroughly expanded to emphasize their potential impact.
  2. The methods section is very detailed, providing an excellent level of transparency for reproducibility. However, the Results section is sometimes overly detailed in describing specific residues, structural shifts, and comparisons. A more concise presentation, perhaps with some of the finer details moved to supplementary information, would improve the readability of the manuscript.

Figures are appropriately referenced and contribute to the clarity of the text. However, the figure captions could be more informative, especially for Figure 1 and the structural comparisons in Figure 3, to help the reader understand what is being depicted and the relevance of the data shown.

  1. The results section, particularly in the structural comparison, provides a wealth of data, but the biological or therapeutic implications of these findings (e.g., potential antibiotic development targeting the MotA1/MotB1 complex) could be more directly tied to the data throughout the manuscript, not just in the Discussion.
  2. The study presents novel structural insights into a chimeric stator complex and discusses potential applications for drug development. However, the conclusion could be strengthened by highlighting more explicitly how these findings contribute to the understanding of flagellar motor function and their broader implications for microbiology and antibiotic design.

Specific Comments

Lines 193-218

  1. The description of the chimeric protein approach is clear, but it would benefit from a more detailed explanation of why this particular chimera was constructed (e.g., what limitations does the native complex pose for functional studies?).
  2. The text mentions “three distinct structures of the complex,” but the rationale behind the differences between these classes could be more clearly stated in this section.
  3. The 3D structure resolution (3.3 Å, 3.3 Å, and 3.5 Å) is a crucial point, but the authors should explicitly discuss how these resolutions impact the interpretation of structural features such as the interface between MotA1 and MotB1.
  4. The description of the A5B2 stoichiometry is essential, but the authors should clarify whether this ratio was expected or unexpected based on previous studies.
  5. The comparison to previously published stator complex structures is useful but could be expanded to more explicitly highlight how the current study’s results differ or provide new insights

 (Lines 222-247)

  1. The interaction between PbMotA1 and EcFliG is clearly described, but a brief explanation of how these interactions drive motor rotation (i.e., the torque generation mechanism) would enhance clarity for the reader.
  2. The RMSD values are provided to demonstrate the structural similarity between PbMotA1 and CsMotA, but their functional significance is not fully explained. The authors should clarify how these similarities impact the functional behavior of the stator complex.
  3. The mention of specific residues (R89, R90, E97, etc.) involved in FliG interactions is important, but these could be shown in a figure or 3D model to make the data more accessible.
  4. The comparison of PbMotA1 to VpPomA is insightful. However, the manuscript would benefit from a discussion of why the observed sequence similarities or differences are important for the function of the flagellar motor.
  5. The functional relevance of the findings should be more strongly emphasized. For example, how do these structural similarities with VpPomA affect our understanding of stator function in Paenibacillus sp. TCA20?

(Lines 254-303)

  1. The structural comparison of the three classes is well-done, but it would be beneficial to include a statement on how these variations relate to the functional states of the motor (e.g., active vs. inactive states).
  2. The text frequently mentions specific structural differences (e.g., the cleft between subunits in Class 2). The significance of these differences could be made clearer by tying them to functional consequences, such as their impact on ion flow or torque generation.
  3. The discussion of LMNG binding and its effects on the structure could be more thoroughly connected to its potential role in disrupting motor function, particularly in relation to antibiotic development.
  4. The RMSD values are again useful, but the authors should more clearly explain what these values imply about the stability of the stator complex or its conformation during motor function.
  5. The section describing the distortion of the MotA1 ring in Class 2 is detailed but could be better linked to the biological function of the stator complex. Does this distortion correlate with a state of motor inhibition or deactivation?

(Lines 312-332)

  1. The description of the plug helix is well-written, but it could benefit from a more direct connection to the functional consequences of its absence in Class 2. How does this absence affect ion transport?
  2. The authors mention that LMNG binding inhibits the binding of the plug helix. It would be helpful to provide a clearer explanation of how this inhibition affects motor function, particularly in terms of torque generation and flagellar rotation.
  3. The comparison of the plug helix in different classes is useful, but the significance of these structural differences for the function of the stator complex could be more explicitly stated.
  4. The finding that one of the plug helices is missing in Class 2 is significant. The authors should discuss whether this represents an intermediate state of the motor or a fully inactive state.
  5. The authors suggest that LMNG might be obstructing ion flow in place of the plug helix. A clearer discussion of how LMNG binding affects the permeability of the ion channel, especially in the context of potential therapeutic applications, would be beneficial.

Lines 337-369

  1. The discussion does a good job of summarizing the key findings, but it could benefit from a more thorough analysis of the functional implications of the structural changes observed, especially in relation to motor activity and antibiotic development.
  2. The authors mention the potential for LMNG to be used as a model for antibiotic design. This is an interesting point that could be expanded upon, particularly by suggesting how similar compounds might be synthesized or tested.
  3. The comparison to previous studies (e.g., Vibrio PomA/PomB with phenamil) is useful, but the authors should more clearly explain how their results differ or extend these findings.
  4. The authors briefly mention “extra densities” found near the α7 helix of MotA1. While speculative, a more detailed discussion of these densities could provide insights into other potential binding sites or structural features of interest.

(Lines 377-387)

The manuscript would benefit from a final statement that ties the findings back to broader questions in bacterial motility and flagellar motor mechanics. How do these results contribute to the general understanding of stator complex function across different species? A more explicit link between the structural findings and their potential application in controlling bacterial motility and infection would enhance the impact of the conclusion.

Suggestions for Supplementary Material

Current Text: “Nine residues (R89, R90, V93, L94, E97, Q113, D117, N233, L235) within 5 Å that potentially interact with FliG were found in two MotA1 subunits…” (Lines 236-237)

Move to Supplementary: A table listing these residues, their positions, and interactions with FliG can be included in the supplementary material.

Current Text: “When the structure of the analyzed PbMotA1 was compared with CsMotA, their root mean square deviation (RMSD) was 1.29 Å, indicating high structural similarity.” (Lines 233-234)

Move to Supplementary: Detailed RMSD comparisons between different classes (Class 1, Class 2, Class 3) and between PbMotA1 and other stator proteins could be moved to supplementary tables or figures.

Current Text: “The α6 helix of chain C was pushed outward by approximately 7.4 Å, and Chain D was shifted about 13 Å towards chain C.” (Lines 287-288)

Move to Supplementary: While these structural shifts are crucial for understanding the effects of LMNG binding, they can be included in supplementary figures or tables with relevant measurements, rather than in the main text, which will make the narrative more concise.

Current Text: “In Class 1, the MotA1 ring has pseudo-pentagonal symmetry. In contrast, in Class 2, there is a large cleft between the neighboring subunits…” (Lines 268-269)

Move to Supplementary: Detailed descriptions of these structural differences, especially the large cleft in Class 2 and the conformation of the MotA1 ring, could be more effectively shown in supplementary figures with annotations rather than being described in full in the main text.

Current Text: “The surface rendering of Class 1 (left) and the cross-section at the blue dotted line…” (Lines 304-305)

Move to Supplementary: More detailed 3D models, structural diagrams, or animations showing the conformational changes and binding sites (e.g., LMNG binding) could be moved to supplementary material to avoid overloading the main section with highly specific structural representations.

Reviewer 2 Report

Comments and Suggestions for Authors

This study reports the structure of the membrane-embedded and cytoplasmic components of the MotAB stator of the flagellar motor of Paenibacillus. The study is technically sound, but the text, and especially the Introduction, is much longer than needed, and important points get lost in the excess verbiage. Also, the suggestion that the study provides a new approach for developing antibacterial agents seems contrived and unlikely to produce anything of practical value.

One of the biggest problems lies in the confusing status of the Paenibacillus flagellar motor. The original report of the isolation of Paenibacillus from a calcium-rich hot spring in 2016 (Imazawa et al., Sci Rep) claimed that motility of this bacterium was driven by a divalent cation current across the cytoplasmic membrane. That was a surprising discovery because hitherto only motility driven by monovalent cations, primarily hydrogen or sodium ions, had been reported. 

In 2020, Onoe et al. (biomolecules) reported that when expressed in E. coli or Bacillus subtilis, the Paenibacillus stator drove flagellar rotation with a hydrogen ion current. That is, motility was driven by the protonmotive force. That study used hybrid stators in which the transmembrane and cytoplasmic components of MotA and MotB came from Paenibacillus. However, the periplasmic or extracellular domain of MotB for E. coli and B. subtilis, respectively, was contributed by the host organism.

It seems highly unlikely that the coupling ion would change when the stator elements that are involved in ion translocation and contact with FliG of the flagellar motor both came from Paenibacillus. A similar construct in which PomA and the transmembrane portion of PomB fused to the PG-binding domain of E. coli MotB are expressed in E. coli provides Na+-driven motility. That is, the coupling ion was determined by the donor of the ion-channel components. By analogy, one would conclude that motility in Paenibacillus is also powered by the protonmotive force.

This discrepancy is never directly addressed. The authors still refer to the calcium-driven motor of Paenibacillus, which seems almost certainly not to exist. The 2020 paper showing that the Paenibacillus motor is very likely proton-driven in Paenibacillus as well should be given thorough coverage in the Introduction of this manuscript. When I read the Abstract before agreeing to carry out the review, I thought I was going to be critiquing a study of the structure the stator unit of a calcium-ion driven motor. However, that turned out not to be the case.

If calcium is required for motility in Paenbcillus, it may well be because of a need for calcium in the general metabolism of that organism. Another, and not exclusive, possibility is that calcium is needed for assembly of the stator unit into the Paenibacillus flagellar motor, something unnecessary when the hybrid MotB is expressed in E. coli.

BL21(DE3) is non-motile and presumably does not make flagellar proteins. Therefore, any stator units produced in that strain will be floating freely in the membrane and be in their inactive state. This is also an important point to state in the Introduction.

The stator units were purified from detergent-solubilized membranes and then incorporated into peptidiscs, from which they were imaged. Therefore, the three classes of structures observed presumably represent an equilibrium distribution of the conformations of the detergent-solubilized stator. The fact that the asymmetric structure with a molecule of the LMNG detergent firmly bound to the periplasmic face of the stator is the form most commonly seen suggests that it is the minimum energy state of the detergent-solubilized stator. This is likely an inactive coonformation, as the authors state, because the ion channels are blocked even though one of the plug helices is displaced. It does make the interesting point that the structure of the MotA pentameric ring is quite flexible, at least in its detergent-solubilized form. That finding is in keeping with the idea that MotA undergoes considerable conformational changes in response to protonation and deprotonation of the critical Asp residue of MotB during the power stroke of the stator.

A crucial point is whether a similar asymmetric structure could exist in the native membrane, perhaps with a phospholipid bound in that position. If so, then perhaps a molecule can be designed that has a high affinity for that conformation of the stator in vivo, that can outcompete phospholipid for binding, and that can lock the stator in an inactive conformation. However, I question whether inhibiting motility is an effective route to take to design an antibacterial agent.

In summary, the data seem solid and the conclusions about the structure are convincing. The paper needs to be rewritten to address the points I raised above. The Introduction should be shortened substantially while providing the important information that is not given. The writing is also stylistically and grammatically substandard, and an effort should be made to improve the quality of the English.

Comments on the Quality of English Language

The English needs a lot of work. As I believe a thorough reworking of the presentation is required, it would be premature to make specific suggestions for improvement of the style or grammar at this time.

Author Response

If calcium is required for motility in Paenbcillus, it may well be because of a need for calcium in the general metabolism of that organism. Another, and not exclusive, possibility is that calcium is needed for assembly of the stator unit into the Paenibacillus flagellar motor, something unnecessary when the hybrid MotB is expressed in E. coli.

Response: Thank you very much for your constructive comments. No information is available about whether the calcium ion is essential for the stator formation and/or assembly in the native cell and stator, although it can be ignored in the expression of the chimeric stator from the E.coli expression system as you described. So following your and other reviewer’s suggestions, we improved the abstract and introduction.

BL21(DE3) is non-motile and presumably does not make flagellar proteins. Therefore, any stator units produced in that strain will be floating freely in the membrane and be in their inactive state. This is also an important point to state in the Introduction.
Response: We added the sentence in the introduction section explaining that the host cell for protein expression did not form the flagellum and the structures we determined are in the inactive state.

A crucial point is whether a similar asymmetric structure could exist in the native membrane, perhaps with a phospholipid bound in that position. If so, then perhaps a molecule can be designed that has a high affinity for that conformation of the stator in vivo, that can outcompete phospholipid for binding, and that can lock the stator in an inactive conformation. 
Response: 
It is quite plausible that, in the native state, phospholipids bind to MotA1/MotB1 instead of LMNG. This is because certain phospholipids have two alkyl chains, which closely resemble the hydrophobic portion of LMNG. However, to date, the structure of the stator complex with bound phospholipids has not been resolved, and there is insufficient information to support this. Therefore, we have added a sentence explaining that the structural similarity between phospholipids and LMNG could aid in the design of novel molecules that inactivate the stator.

The English needs a lot of work. As I believe a thorough reworking of the presentation is required, it would be premature to make specific suggestions for improvement of the style or grammar at this time.
Response: We utilized MDPI English Editing to enhance the language quality of our manuscript.

Round 2

Reviewer 1 Report

Comments and Suggestions for Authors

comments addressed by the authors. 

Author Response

Thank you for your review.

Your advice helped improve the readability and made the paper much better.

Reviewer 2 Report

Comments and Suggestions for Authors

The authors have done an excellent job of responding to my critique of the original submission. Experiences like this increase my enthusiasm for taking on review assignments.

Relatively minor changes to the English presentation are selected below. I have not made a specific suggestion about concerns over paragraph length. Any paragraph that is too long becomes hard to follow just because of its length.

Comments on the Quality of English Language

The most important suggestion is to go through the entire manuscript and break up the paragraphs that extend over more than ten lines into several shorter paragraphs. I am sure the authors can find logical breaks in their narrative that can indicate where a new paragraph should be started.

Minor changes:

Line 74. "In this study, ..."

Line 198. "...MotB (residues 104-302)...

Line 220. Should be CsMotA.

Line 226. Rewrite as "...of FliG that can potentially interact with it."

Line 229. Replace "homology" with "similarity."

Line 243. Should be singular "structure." 

Line 261. Replace "the" with "two."

Line 303. rewrite as "...emphasized; the rest are...."

Line 308. the "weare" should be "were."

Line 309. Delete period. "...subunits (between...."

Line 347. The "that" should be "which."

Lines 373 to 377. Rewrite as " There is a technical hurdle in that an in vitro assay for the measurement of the coupling ion flux of the stator has not been established yet, and solving this problem requires a further study. But by combining such a new assay with the screening of small molecules designed based on the structure of the LMNG, new drugs may be developed to serve as antibiotics with a much broader antimicrobial spectrum than those presently available."

Lines 388-389. Delete "stator complex."

Line 392. Replace "to" with "with" and delete "disruptive."

Line 397. Rewrite as "...molecular motors and are expected

Author Response

We've divided the long paragraphs in the results section as you suggested.

Also, we've implemented all of your proposed edits to the manuscript. 

We are grateful for your review.

Your insightful advice contributed to the enhanced readability and overall improvement of the paper.